# Remittances and Household Expenditure in Nepal: Evidence from Cross-Section Data

**Sridhar Thapa [1],\* and Sanjaya Acharya [2]**

1   UN World Food Programme, Patandhoka Road, Chakupat, Lalitpur 10, Nepal
2   Saraswati Multiple Campus, Tribhuvan University, Thamel 26, Kathmandu; sanjaya.acharya@gmail.com
\*   Correspondence: sridhar.thapa@wfp.org; Tel.: +977-1-5260607

**Abstract:** This paper examines the effect of remittances on household expenditure patterns applying propensity score matching methods that allow designing and analyzing observational data and enable reducing selection bias. We use data from the Nepal Living Standards Survey 2010/2011. In general, remittance recipient households tend to spend more on consumption, health and education as compared to remittance non-receiving households. Although the findings do not clearly provide evidence of either the productive or non-productive use of remittances, expenditures on non-food investment categories, such as durable goods, health and education, are more apparent among remittance-receiving households compared to remittance non-receiving households, which signal the prospect of a sustainable long-term welfare gain among the former.

**Keywords:** remittances; household expenditure; Nepal; propensity score matching

## 1. Introduction

Remittances, defined as financial inflow arising from the cross-border movement of nationals of a country, are the transfer of money and goods sent by migrant workers to their country of origin. Globally, remittance flows to low and middle income countries (LMICs) are estimated to be a total of UD$ 442 billion in 2016, an increase of 0.8 percent over the past year, in which the officially-recorded remittance receipt developing countries are mostly from Asia, Latin America, Eastern Europe and Africa (World Bank 2016). The amount of remittance sent by migrants to developing countries is about three-times higher than the official development assistance and can play a significant role in the overall development and human welfare in the recipient countries. It is also estimated that remittances through informal channels could add at least 50 percent to the globally-recorded flows (World Bank 2006).

The migration from Nepal, in addition to India, to the Middle-East (Saudi Arabia, Qatar and United Arab Emirates) and Southeast Asia, such as Malaysia[1] had not only dramatically increased during the armed conflict period in Nepal, but also prolonged for a decade even after signing the Comprehensive Peace Agreement in 2006. It made the absentee population increase by about two million, more than double that of the year 2001 (Central Bureau of Statistics 2012).

In the situation of prolonged and sluggish economic growth during conflict and transition in Nepal, remittances are significantly contributing to reduce poverty, household income and

---

1   Nepal Migration Survey 2009 shows that remittances from Gulf countries account about 49% followed by India (19%) and Malaysia (10%)—major destinations for Nepali migrant workers. However, India is most preferred destination for many rural people due to neighboring country and no visa requirement with low migration cost.

expenditure. Moreover, it has become the single biggest source of foreign exchange earnings in Nepal. Currently, Nepal is the first remittance-receiving country in the world in terms of the percentage of the Gross Domestic Product (GDP); this swelled from 1.5 percent of GDP in 1993 to 32.2 percent in 2015—6.7 billion US$ (World Bank 2016). It has profound implications for the country's economy. For instance, the Nepal Living Standard Surveys (NLSSs) show that remittance income has significantly contributed reducing the head count poverty rate from 42 percent in 1995/1996 to 31 percent in 2003/2004 and then 25 percent in 2010/2011, as the percent of sampled households receiving remittances has sharply increased from 23.4 percent in 1995/1996 to 55.8 percent in 2010/2011 (Central Bureau of Statistics 2011). Moreover, remittance flow in Nepal has offset large trade deficits and enabled maintaining a current account surplus (Mohapatra et al. 2010). Despite a significant contribution to both the micro- and macro-economy of the country, relatively few studies have assessed the impact of remittances on the household and national economy. Therefore, an understanding of the use of remittances and expenditure patterns at the household level can have significant policy implications in the host country.

The migration and consequent remittances have become a virgin area of research for the past couple of decades due to the growing number of migrants to abroad and the swelling of the remittance flowing to developing countries across the world. Moreover, the study of the economic impact of remittances on the recipient households can have particular interest among policy makers and researchers, especially on how these remittances are spent or used and what the effects of remittances are on the expenditure patterns of the recipient households. Moreover, the study of the comparative analysis of the expenditure pattern of remittance-receiving and non-receiving households would be even more innovative for drawing insights for more productive use of the remittances.

The use of remittances is broadly studied from three different perspectives (Adams and Cuecuecha 2010). The first one, probably the most widespread, is considered pessimistic, which considers the expenditure pattern of remittance income quite similar to other regular household incomes, such as farm and non-farm incomes of rural households. For instance, Thapa (2009) found remittance income as a substitute of non-farm income. The second view considers remittance income leading to a behavioral change among the recipient households mainly due to the change in consumption pattern rather than going to investment. For example, Chami et al. (2003) present a good example of this pattern. The third view is derived from the permanent income hypothesis, assuming that since remittances are a transitory type of income, households spend more at the margin on investment goods such as human and physical capital investments than on consumption goods, which can contribute positively to economic development as explored by Adams (1998). Some studies, in line with the permanent income hypothesis, have shown a large positive impact on student retention rates in El Salvadorian school (Edwards and Ureta 2003), a significant increase in educational expenditures in the Philippines as a result of positive exchange rate shocks (Yang 2008) and investment in housing in Nigeria (Osili 2004). Likewise, Adams and Cuecuecha (2010) found a positive impact of remittances on the margin of two investment goods, education and housing, in Guatemala. Furthermore, Adams (1998) shows a significant and positive impact of internal and external remittances on two types of physical asset accumulation: irrigated and rainfed land in rural Pakistan. The study by Quisumbing and McNiven (2010) showed a positive impact of remittances on housing, consumer durables and non-land assets in rural Philippines. Cuong (2009), however, investigates the impact of internal and international remittances on the household welfare in Viet Nam using household living standard survey data of 2002 and 2004. The author found a greater impact of remittances on non-food expenditures as compared to food expenditure, while international remittances were largely used for savings and investment.

In Nepal, Bohra-Mishra (2013) investigated the effects of labor migration on different types of investments in Chitwan district and found a positive role of labor migration and likelihood of spending on agriculture, relatively productive investment. Regmi and Tisdell (2002) showed a large spending of remittance income received from rural-to-urban migration on household expenditures, mainly in education for their close relatives, and other expenses, such as in agriculture. Likewise, using NLSS

1995/1996 data, Bansak and Chezum (2009) found a positive impact of remittances on household expenditure on the education of children, the human capital formation of school age boys and girls. Vogel and Korinek (2012) also support this finding using NLSS 2003/2004 data.

This study aims to shed light on the effect of remittances on the expenditure patterns of the recipient households; however, contrary to previous studies, it assesses the differential impacts of internal, external and the combination of both remittances on the household expenditure patterns. Households receiving external remittances are, on average, relatively richer families as compared to households receiving internal remittances. Moreover, per capita internal remittance received by poorer households is relatively less than per capita external remittance among the recipient households; consequently, the former has a role in consumption smoothing that may lead to reduce poverty and hunger. On the contrary, per capita external remittance is relatively high with likely expenditure on accumulating durable goods (e.g., kitchen appliances, jewelry, furniture, motorcycle, car, electronic goods, etc.), healthcare and education. It is also reasonable to assume that internal migration is less risky compared to international migration, and remittance received from the former can provide insurance to poorer households who often migrate from rural to urban areas for employment opportunities. Studies exist that focus on the impact of remittances on particular household expenses/consumption by product type; however, a study that makes comparative analysis of expenditure patterns among food, education, health, durable goods and other expenses such as on rent, tobacco, electricity, fuels, and so on, in a single framework is virtually absent. This gap the present study intends to fulfil. Moreover, this study uses the latest NLSS data, nationally representative survey data for 2010/2011, and analyzes the impacts with reference to different income groups and also by ecological regions.

The rest of the paper is organized as follows: Section 2 presents the empirical approach, mainly the choice of functional relationship between remittance income and household expenditure. Data and descriptive statistics are presented in Section 3, followed by the results of the models in Section 4. Section 5 concludes the paper.

## 2. Empirical Approach

One of the major objectives of this study is to assess whether remittance-receiving households have higher expenditures on different food and non-food items, the choice of functional form does matter in order to capture the impact of remittances. Many studies have applied an Engel curve approach often analyzed in the Working–Leser model (Tabuga 2007; Valero-Gil 2009; Adams and Cuecuecha 2010; Gobel 2013), in which the model assumes budget shares following a linear relation to the logarithm of the total expenditure. Despite the advantages of adding up restriction in the Working–Leser method[2], the model often encounters the endogeneity problem of the remittance variable linked to the reverse causality between household expenditure patterns and remittance recipients (Clement 2011). The use of an instrumental variable (IV) approach can be a solution to this problem. However, in the IV approach, finding good instrumental variables is always challenging, which can lead to large inconsistencies in the IV estimate. For example, McKenzie and Sasin (2007) and Adams (2011) tested different instrument variables such as distance between the country of origin and the country of migration, migration level (at a village or community level) and access to transport infrastructure and financial institutions, but none of these variables fit perfectly well to relate migration and remittances. This paper, therefore, uses an alternative approach, a propensity score matching (PSM), to fix the potential bias in assessing the effect of remittances on expenditure patterns, previously applied by Esquivel and Huerta-Pineda (2007) and Clement (2011).

Propensity score matching (PSM), first proposed by Rosenbaum and Rubin (1983), is an alternative approach to estimate the effect of receiving treatment when random assignment of treatments to

---

[2]　This is the case when budget share of one commodity increases, share of another commodity must be reduced due to the budget constraint of the household (See, Clement 2011)

subjects is not feasible. The PSM method reduces the bias in the estimation of treatment effects with observational datasets and is now becoming increasingly popular in the evaluation of economic policy interventions (Becker and Ichino 2002) and also in studying the nexus among migration, remittance and household expenditure (Esquivel and Huerta-Pineda 2007; Clement 2011; Bohra-Mishra 2013). The study shows that propensity score methods perform better for larger sample sizes (e.g., more than 1000 observations) (Zhao 2004).

The propensity score is the probability of receiving treatment (in this case remittance), conditional on the covariates, and is often applied to correct the estimation of treatment effects controlling the existence of these confounding factors based on the idea that the bias is reduced when the comparison of outcomes is performed using treated and control subjects (Becker and Ichino 2002). The model is estimated in two stages: in the first stage, estimate logit or probit models whether households receive remittance as binary dependent variables with other socioeconomic variables that affect receiving remittances as independent variables, and then, find the propensity score, which is the probability of receiving remittances conditional to the characteristics included in the regression model. In the second stage, we look at the effect of treatment on the outcome matching remittance receiving with non-receiving households. Finally, households are matched with similar households and all unmatched units discarded (Rubin 2001). However, PSM is primarily applied for two groups under consideration and cannot be applied to analyze more than two groups.

As discussed earlier, the propensity score is the conditional probability of receiving a treatment given pre-treatment characteristics. $R_i$ ($i = 1$ and 0) is the binary variable equal to 1 when the subject of treatment (i.e., household receiving remittances) and otherwise 0. Suppose that $Y_{1i}$ is the value of the variable of interest (expenditure patterns as outcome variables) and $Y_{0i}$ is the value of the same variable when households do not receive any remittance (0). Let $\tau_i$ be the treatment effect for a single unit, $\tau_i = Y_{i1} - Y_{i0}$, then the treatment effect with and without relevant outcome indicators can be given by:

$$\tau = E[Y_{i1}|R_i] - E[Y_{i0}|R_i = 1] \tag{1}$$

where $\tau$ is the average treatment effect (e.g., average difference between the treated households: i.e., remittance receiving households and non-treated households, i.e., households without remittances). This is a simple comparison between remittance receiving and non-receiving households that may not provide the actual impact of the treatment as we intend to select in treatment. It is also a fact that other factors correlated with the treatment and omitted variable can also affect the outcome variables because of the problem of observing both treatment and control at the same time.

However, propensity score matching methods, which are based on the conditional independence assumption (CIA), show that the potential outcomes are independent of the treatment status, denoted by $X$ (Rosenbaum and Rubin 1983):

$$[Y_{i1}, Y_{i0}] \perp R_i | X_i \tag{2}$$

where, given $X_i$, the potential outcomes are independent of the treatment status, or after controlling for $X$, the treatment assignment is as good as random. This assumption is also called unconfoundedness or selection on observables.

$$E[Y_{i0}|R_i = 1, X_i] = E[Y_{i0}|R_i = 0, X_i] \tag{3}$$

Here, it is possible to participate conditionally in propensity score matching denoted by $P(X)$ rather than on observable characteristics $X$. The propensity score is the probability of taking the treatment given a condition of the vector of observed variables.

$$P(X_i) = \Pr[R_i = 1|X_i] \tag{4}$$

Counterfactual estimation can be presented as:

$$E[Y_{i0}|R_i = 1, P(X_i)] = E[Y_{i0}|R_i = 0, P(X_i)] \tag{5}$$

Therefore, the average treatment effect for individual *i* can be measured by:

$$ATT = E[Y_{i0}|R_i = 1, P(X_i)] - E[Y_{i0}|R_i = 0, P(X_i)] \tag{6}$$

After estimating propensity scores derived from logit or probit, there is a need to select a suitable matching estimator in order to compare units related to treated units.

$$\widehat{ATT} = \frac{1}{T} \sum_{i=1}^{T} \left[ Y_{i1} - \sum_{j=1}^{N} W(i, j) Y_{ij0} \right] \tag{7}$$

Here, $Y_{i1}$ is the post-treatment outcome of treated unit *i*, $Y_{ij0}$ is the outcome of the *j*th non-treated unit matched to the *i*th treated unit, *T* refers to the total number of treated units, while *N* means the non-treated units, and the positive valued weight function is measured by $W(i, j)$. There are several matching techniques applied to match households or individuals based on the propensity scores; this paper uses the kernel estimator, as it matches each treated unit to a weighted sum of comparison, with the greatest weight assigned to units with closer scores (Heckman et al. 1998). This can be given as:

$$W(i, j) = \frac{K\left(\frac{p_i - p_j}{h}\right)}{\sum_{j \in \{R=0\}} K\left(\frac{p_i - p_j}{h}\right)} \tag{8}$$

where $p_i$ refers to the propensity score of treated unit *i*, $p_j$ is the propensity score comparison unit *j* and *h* is a bandwidth parameter. This is a consistent estimator of the counterfactual outcome $Y_{0i}$. Kernel matching can be used either for all comparison units (e.g., Gaussian kernel) or propensity scores with a fixed bandwidth. In this paper, we use the Gaussian kernel estimator, which has the advantage of making maximum use of all of the observations. Kernel estimation has the advantage of addressing selection bias because it computes the effect of treatment for a particular treated observation as a weighted average of its difference outcome from all of the untreated observations. It also resembles random assignment no more than any other non-experimental methods.

## 3. Data and Descriptive Statistics

The study uses the data from the Nepal Living Standard Survey (NLSS) 2010/2011 conducted by the Central Bureau of Statistics, Nepal, with financial and technical support from the World Bank. This is the third national living standard survey that makes a periodic update of income, poverty measurement and household consumption behavior, including food security and nutrition. The NLSS III applied the methodology developed and promoted by the World Bank, and the survey covered all ecological belts, ethnic groups and income groups across the country, with 5988 household samples from 499 primary sampling units (PSUs). The PSUs were selected with the probability proportional to size. The NLSS captures the information of housing, access to facilities, migration, remittances and social transfer, household income and expenditure, consumption, agriculture and livestock and other demographic and socioeconomic characteristics.

### 3.1. Remittances

The information on the descriptive statistics includes household characteristics such as proportion of infants, children and elders in the household size, regional dummies, age, sex, education and occupation of household head, the dependency ratio, the percentage of migrants from the primary sampling unit (PSU) including the share of expenditure of different food and non-food items in both remittance recipient and non-recipient households. A migrant is considered as an individual who has left the household/family for at least six months to live or work elsewhere, either within the country or abroad. Table 1 presents the descriptive statistics of the variables used in the model. A remittance-receiving household for this study is the household that received some remittance

inflow either from absentees or from relatives and siblings within the last 12 months. Out of the total remittance recipient households, those receiving higher remittances are from external migrants (37%) compared to inter-migrants (23%). Likewise, about 50 percent of the households with its head in agriculture receive remittances compared with other occupation households. Likewise, illiterate heads of household receive a relatively high percentage of remittances (49%) compared to literate or educated household heads. Out of total remittance recipient households, a significantly higher number of rural households (71%) receive remittances compared to urban dwellers (29%). In terms of ecological belt, remittance recipient in the total remittance recipient households are relatively high in the hill (47%) and Tarai (46%), the southern plain, as compared to mountains.

**Table 1.** Descriptive statistics.

| Variables | Type of Variable | Remittance Recipients | Remittance Non-Recipients | All |
|---|---|---|---|---|
| Share of children < 6 years | Continuous | 0.13 (0.16) | 0.12 (0.15) | 0.13 (0.16) |
| Share of children 7–15 years | Continuous | 0.21 (0.21) | 0.21 (0.20) | 0.21 (0.21) |
| Share of elders (65+) | Continuous | 0.08 (0.18) | 0.07 (0.18) | 0.07 (0.18) |
| Household size | Continuous | 4.61 (2.4) | 4.94 (2.19) | 4.75 (2.31) |
| Dependent ratio | Continuous | 0.40 (0.29) | 0.45 (0.26) | 0.42 (0.28) |
| Migration from the cluster (%) | Continuous | 6.81(1.76) | 6.45 (1.53) | 6.65 (1.67) |
| Rural | Dummy | 0.71 (0.45) | 0.59 (0.49) | 0.65 (0.47) |
| Urban | Dummy | 0.29 (0.44) | 0.41 (0.49) | 0.35 (0.47) |
| Hill | Dummy | 0.47 (0.49) | 0.61 (0.48) | 0.53 (0.50) |
| Tarai | Dummy | 0.46 (0.49) | 0.32 (0.46) | 0.39 (0.49) |
| Mountain | Dummy | 0.07 (0.26) | 0.06 (0.24) | 0.07 (0.25) |
| Female to male ratio | Continuous | 1.01 (0.78) | 1.04 (0.66) | 1.03 (0.73) |
| Age of household (HH) head | Continuous | 46.2 (14.79) | 44.78 (13.35) | 45.99 (14.32) |
| Illiterate | Dummy | 0.49 (0.50) | 0.39 (0.48) | 0.44 (0.49) |
| Primary | Dummy | 0.13 (0.33) | 0.15 (0.35) | 0.14 (0.34) |
| Secondary | Dummy | 0.12 (0.33) | 0.13(0.35) | 0.13 (0.33) |
| High School | Dummy | 0.10 (0.29) | 0.13(0.34) | 0.11 (0.32) |
| University (+11) | Dummy | 0.16 (0.36) | 0.19(0.40) | 0.17 (0.38) |
| Head occupation, agriculture | Dummy | 0.50 (0.50) | 0.37 (0.48) | 0.44 (0.49) |
| Head occupation, non-agriculture | Dummy | 0.36 (0.48) | 0.53 (0.40) | 0.44 (0.49) |
| Head occupation, unemployed/not reported | Dummy | 0.14 (0.33) | 0.10 (0.29) | 0.12 (0.31) |
| Share of food in the total expenditure | Continuous | 0.58 (0.17) | 0.57 (0.18) | 0.57 (0.16) |
| Share of non-food in the total expenditure | Continuous | 0.14 (0.07) | 0.13 (0.07) | 0.14 (0.07) |
| Share of education in the total expenditure | Continuous | 0.07 (0.09) | 0.07 (0.08) | 0.07 (0.06) |
| Share of health in the total expenditure | Continuous | 0.06 (0.12) | 0.05 (0.10) | 0.05 (0.11) |
| Share of durable goods in the total expenditure | Continuous | 0.03 (0.04) | 0.04(0.06) | 0.03 (0.05) |
| Share of other items in the total expenditure | Continuous | 0.15(0.11) | 0.16 (0.12) | 0.15 (0.10) |
| Remittance recipient households (HHs) | Dummy | 1(0) | 0 | 0.53 (0.50) |
| Internal remittance recipient HHs | Dummy | 0.23 (0.42) | 0 | 0.18 (0.39) |
| External remittance recipient HHs | Dummy | 0.37 (0.48) | 0 | 0.24 (0.43) |
| Both internal and external remittance recipient HHs | Dummy | 0.11 (0.32) | 0 | 0.07 (0.26) |
| **Observations** | | **3178** | **2810** | **5988** |

Note: values are the mean and standard deviation in parenthesis.

Table 2 presents the status of remittance inflow from internal, external and both internal and external sources and its distribution across the ecological belt and per capita expenditure by quintile group. The data show that out of 5988 sample households, about 53 percent received remittances, 18.6 percent internally, 24.5 percent externally and 7.1 percent from both internal and external sources. About 57 percent of rural households receive remittances as compared to 44 percent of urban households. Tarai belt (the southern plain area) has the highest percent of remittance-receiving households (61.7%) in the total sampled households, followed by the mountain (56.1%) and hill (46.3%) belts. The higher is the per capita expenditure by quintile group, the higher is the level of remittances.

**Table 2.** Descriptive statistics of per capita remittance income (in NPR).

| | Total Remittances | | Domestic Remittances | | Foreign Remittances | | Both-Domestic and Foreign Remittances | |
|---|---|---|---|---|---|---|---|---|
| | Remittance Receiving HHs | Remittance pc Income | Remittance Receiving HHs | Remittance pc Income | Remittance Receiving HHs | Remittance pc Income | Remittance Receiving HHs | Remittance pc Income |
| All | 53.07% | 25,323.34 | 18.65% | 16,160.29 | 24. 53% | 28,552.51 | 7.06% | 38,351.34 |
| Rural | 57.59% | 21,041.88 | 12.31% | 16,068.14 | 19.54% | 20,929.73 | 7.4% | 33,949.72 |
| Urban | 44.64% | 36,778.98 | 21.38% | 16,424.29 | 29.38% | 46,685.22 | 3.4% | 53,548.50 |
| | | | | Ecological Belt | | | | |
| Mountain | 56.13% | 11,820.52 | 26.72% | 8541.74 | 20.83% | 10,738.38 | 7.6% | 24,641.74 |
| Hill | 46.29% | 26,238.82 | 16.45% | 10,699.99 | 23.60% | 34,283.94 | 5.8% | 39,352.97 |
| Tarai | 61.70% | 26,609.03 | 19.15% | 25,407.92 | 30.01% | 24,102.56 | 6.1% | 40,002.74 |
| | | | | Income Quintile | | | | |
| 1st | 44.59% | 5964.47 | 10.48% | 2824.36 | 23.85% | 6588.34 | 4.4% | 10,650.45 |
| 2nd | 49.50% | 8890.48 | 13.40% | 3892.42 | 24.80% | 11,602.96 | 4.4% | 97,87.42 |
| 3rd | 53.48% | 15,669.47 | 19.29% | 7504.60 | 26.25% | 19,904.32 | 7.8% | 21,738.25 |
| 4th | 55.00% | 20,016.79 | 18.46% | 8170.93 | 26.69% | 28,980.87 | 6.7% | 23,300.57 |
| 5th | 57.76% | 49,716.32 | 24.12% | 33,655.77 | 26.94% | 58,066.80 | 9.8% | 68,675.90 |

Note: pc = per capita, HHs = households. Calculation based on data from the Nepal Living Standard Survey (NLSS) 2010/2011, Central Bureau of Statistics, Kathmandu.

Per capita remittance income is high among the households receiving both internal and external remittances. Overall, urban households have higher per capita remittance income. In general, households having both internal and external migrants received higher remittance income compared to migrants having only internal or external migrants. The highest per capita remittance income both from internal and external sources is apparent among urban and Tarai households; whereas the highest per capita external remittance income is among the hill households and internal remittance among the Tarai households. Per capita remittance income is more than six-fold in the highest expenditure quintile than in the poorest expenditure quintile in both internal and external remittances; while this is more than 10-times higher in internal and total remittances (Table 2).

### 3.2. Household Expenditure Patterns

Overall, the household expenditure share for food items is 58 percent followed by 14 percent in non-food daily consumption goods. The shares of long-term expense items such as education, health and durable goods account for 6, 6 and 3 percent, respectively; whereas 16 percent of the total expenses are unclassified (Table 3).

**Table 3.** Average budget share of both households.

|  | Household with Remittances | Household without Remittances | Difference | Two-Way *t*-Test (*t*-Statistics) |
|---|---|---|---|---|
| Food | 0.58 | 0.57 | 0.01 | 2.38 ** |
| Non-food | 0.14 | 0.13 | 0.01 | 3.65 *** |
| Education | 0.068 | 0.067 | 0.001 | 0.46 |
| Health | 0.06 | 0.05 | 0.01 | 3.59 *** |
| Durable goods | 0.036 | 0.037 | −0.001 | 1.02 |
| Others | 0.16 | 0.15 | 0.01 | 5.2 *** |

Note: ** and *** refer to significant at the 5% and 1% levels, respectively. Durable goods are mostly fixed assets, such as television, interior decoration goods (e.g., furniture, pillows, mattresses, blankets, etc.), motor cycle, car, radio, electronic goods, and so on; while others include expenditure on tobacco, water, electricity rent and garbage.

The average budget shares of remittance-receiving and non-receiving households show a marginal difference. For instance, remittance-receiving households tend to spend more on non-food consumption, education and health, while remittance non-receiving households spend more on the consumption of food and durable goods. However, there is no significant difference in the proportion of expenditure on durable goods. These results suggest that migration and remittances seem to be spent more on human capital development and health, together with non-food goods.

## 4. Empirical Results

### 4.1. Motivation to Remit

This study proceeds with applying the logit model to estimate the probability of receiving remittances. More specifically, we estimate the logit model with the probability of receiving remittances versus no remittances, internal remittances versus no remittances, external remittances versus no remittances as binary dependent variables with a number of explanatory variables (e.g., socioeconomic and household characteristics including geographical dummies), assuming that these variables do matter for the motivation to remit. As the propensity score is the predicted probability of treatment derived from the logistic regression model, we need to pay attention whether the explanatory variables used for the logit model can explain the remittance flows. More importantly, covariates should have been chosen in such a way that they can avoid reverse causality with remittance recipients because household expenditure patterns should not be affected by the explanatory variables other than through remittance channels (Clement 2011). In order to address these issues, variables included as household characteristics, such as the proportion of children equal to or less than six years, between seven and

15 years and elders (65+), household size with migrant members and dependents ratio[3], female to male ratio, household head's age, education and occupational status and place of residency whether in rural or urban areas and in mountain or hill or Tarai belts, are considered as covariates that influence remittance receipts, but not the expenditure patterns. We also include the percentage of migrant people out of the total sample population in the primary sampling unit (PSU) as an explanatory variable because of the strong positive effect of the network in the migration decision (Thapa 2009). Furthermore, this is also because of receiving remittance mainly in the situation where the informal source of receiving remittance is the major source of household income for those families who have migrant worker(s) in India and send remittances through their relatives and neighbors (World Bank 2011).

Table 4 presents the results of the logit regressions to estimate the propensity scores for total, internal, external and both internal and external remittances. The models run with a number of explanatory variables mentioned above from a sample of 5988 households, and the results showed a good explanatory power, as these variables explain 65.5 percent, 81 percent, 76 percent and 93 percent of the variations in total, internal, external and both internal and external remittances, respectively. McFadden's pseudo $R^2$ is found to be relatively high in both internal and external remittances (13%) followed by external (9%) as compared to total remittances (7%) and internal remittances (7%), implying that the explanatory power is relatively high in external and both internal and external remittances compared to total remittances and internal remittances.

**Table 4.** Logit estimates for remittance recipients.

| | Total Remittances | | Internal Remittances | | External Remittances | | Both External and Internal Remittances | |
|---|---|---|---|---|---|---|---|---|
| | Coefficient | Standard Error | Coefficient | Standard Error | Coefficient | Standard Error | Coefficient | Standard Error |
| Household characteristics | | | | | | | | |
| Share of children < 6 years | 0.84 *** | 0.22 | 0.19 | 0.29 | 1.62 *** | 0.27 | 1.01 ** | 0.45 |
| Share of children 7–15 years | 0.36 ** | 0.16 | −0.06 | 0.22 | 0.98 *** | 0.21 | −0.82 *** | 0.36 |
| Share of elders (65+) | 0.10 | 0.19 | −0.89 *** | 0.23 | −0.98 *** | 0.25 | −1.44 *** | 0.40 |
| Household size | −0.14 *** | 0.02 | −0.21 *** | 0.02 | −0.16 *** | 0.02 | −0.15 *** | 0.03 |
| Dependent ratio | −0.72 *** | 0.12 | −0.17 | 0.15 | −0.83 *** | 0.14 | 0.15 | 0.22 |
| Migration from psu (%) | 0.11 *** | 0.02 | −0.02 | 0.02 | 0.19 *** | 0.02 | 0.17 *** | 0.03 |
| Rural dummy | 0.29 *** | 0.07 | 0.71 *** | 0.09 | 0.15 * | 0.09 | 0.50 *** | 0.15 |
| Hill | −0.38 *** | 0.11 | −0.53 *** | 0.13 | 0.26 * | 0.15 | −0.20 | 0.21 |
| Tarai | 0.27 ** | 0.11 | −0.42 *** | 0.14 | 0.52 *** | 0.16 | −0.18 | 0.22 |
| Female to male ratio | 0.08 ** | 0.04 | 0.17 *** | 0.06 | 0.13 ** | 0.05 | 0.33 *** | 0.08 |
| Age of HH head | 0.02 | 0.02 | 0.02 *** | 0.003 | 0.01 *** | 0.003 | 0.04 *** | 0.01 |
| Education of HH head | | | | | | | | |
| Primary | −0.18 ** | 0.09 | −0.13 | 0.11 | −0.39 *** | 0.11 | −0.17 | 0.17 |
| Secondary | −0.05 | 0.09 | 0.12 | 0.12 | −0.43 *** | 0.11 | 0.38 ** | 0.17 |
| High School | −0.32 | 0.1 | 0.01 | 0.13 | −0.67 *** | 0.13 | −0.25 | 0.20 |
| University (+11) | −0.05 | 0.09 | 0.24 ** | 0.11 | −0.52 *** | 0.11 | −0.25 | 0.20 |
| Occupation | | | | | | | | |
| Agriculture | 0.04 | 0.10 | 0.30 ** | 0.13 | 0.15 | 0.13 | 0.41 ** | 0.19 |
| Non-Agriculture | −0.35 *** | 0.10 | −0.19 | 0.13 | −0.48 *** | 0.12 | −0.38 | 0.20 |
| Constant | 0.34 | 0.23 | −0.90 *** | 0.31 | −2.01 *** | 0.32 | −4.87 *** | 0.48 |
| Observations | 5988 | | 4102 | | 4448 | | 3408 | |
| Pseudo R-sqr McFadden | 0.07 | | 0.07 | | 0.09 | | 0.13 | |
| Percent of correct | 65.5% | | 81.5% | | 76.0% | | 92.9% | |
| LR test | 539.86 *** | | 299.78 *** | | 504.57 *** | | 329.63 *** | |

Note: psu = primary sampling unit, HH = household. *, ** and *** are level of significance at 10%, 5% and 1%, respectively. Sampling weights are not used to calculate the propensity score.

The probability of receiving remittances is positive and significant if the household has a higher share of children below or equal to six years of age, implying that highly dependent family members in the households might require additional income from remittances for child education. However, on the contrary, the probability of receiving remittances is likely to be low along with the increase in the number of elders in the family and a larger family size. The probability of receiving remittances

---

[3]  It is the ratio of population of aged bellow 14 years (young population) and those 60 years and above (older population) to the population of the productive age groups (15–59 years).

becomes positive and significant along with the rise in the ratio of females in the family, implying that most of the migrants are male and they send remittances to their female counterparts in order to maintain daily household expenses, child education and healthcare. Receiving remittance is expected to be positively correlated with the age of the household head since younger men migrate more often than older people and the latter are the major remittances recipients. Likewise, rural households have a large number of migrants and are more likely to remit more than urban people, suggesting that social networks are stronger in rural areas, which eventually helps to raise the remittance income.

The probability of receiving total and external remittances is significant and positive if the households are from the Tarai belt, perhaps due to more access to India and abroad coupled with relatively good road networks in Tarai. However, the probability of receiving remittances in the hills seems to be negative from total and internal sources, but positive for external remittances. Education level of the household head does not seem to be supportive in this regard if the household has external migrants; however, the probability of receiving internal remittance is positive if the household head has a college level education. As many educated people go to urban areas or other districts for work, they are likely to send remittances to their family at home. The probability of receiving remittances is negative and significant for those households engaged in non-agricultural activities such as services, business and entrepreneurship. Receiving internal and both internal and external remittance is positive and significant while the households are mostly involved in subsistence agriculture. Remittance income through migrant members seems to be supportive for households mostly living in rural areas and relying on subsistence agriculture for their livelihoods.

### 4.2. Impact of Remittances on Household Expenditure

In order to estimate the impact of remittance on household expenditure or on the outcome variables, we first estimate logit models to derive the propensity scores, and then, we use average treatment effects using Gaussian kernel matching for total, internal, external and both internal and external remittances combined. Gaussian kernel matching can be more efficient, and it allows using all of the data from the untreated groups. Kernel matching seems to be more consistent and efficient when observations to be compared are large and asymptotically distributed.

We conducted a couple of diagnostic tests to assess the performance of the variables before and after matching. The first one is the balancing test proposed by Rosenbaum and Rubin (1983) and applied in Dehejia and Wahba (2002), which is the *t*-tests for the equality of means in the treated and control groups before and after matching. For good balancing, there should be a non-significant difference after matching. Second is the standardized bias before and after matching, which is the percentage difference of the sample means in the treated and non-treated subsamples as a percentage of the square root of the average sample variance in both groups. A bias reduction in most empirical studies of less than five percent is considered as sufficient (Caliendo and Kopeinig 2008). The third one is sensitivity analysis using Rosenbaum bounds to assess the effects of remittances on several expenditure groups as outcome variables. The results show that the *t*-values of most variables after matching are non-significant in total, internal, external and both internal and external remittances, and the percentage of bias is also less than five percent in most of the variables with the exception of a few, such as children under six years, children between seven and 15 years and the sex of the household head. This shows a good balance between the treated and control groups. Moreover, the percentage of bias is around 10 percent for most expenditure shares of food and non-food items. However, the proportion of bias reduction for each covariate is found to be greater than 60 percent for total, internal, external and both internal and external remittances, implying that the covariates are well balanced and the matching was effective in building a good control group. This study has also carried out sensitivity analysis using Rosenbaum bounds for the effects of remittances on the share of expenditures in different groups as outcome variables (see Table 5). The critical value of $\Gamma$ (gamma) ranges from 1.05 to 2.00, which mostly seems to be in the range between 1.1 and 2.0, as shown in the majority of the literature (Bertoli and Marchetta 2014; Clement 2011). However, the critical value

of $\Gamma$ varies significantly between the Hodges–Lehmann point estimates at the 95 percent confidence interval. For instance, the lowest critical value of food consumption is 1.05 at the 95 percent confidence interval, implying that the effects of remittances on this outcome are more sensitive to hidden bias, while the effects of remittances on health expenditure seem to be less sensitive to hidden bias as the critical value of $\Gamma$ ranges from 1.8 to 2.00.

**Table 5.** Average treatment effects (total remittances) and results of the sensitivity analysis.

|  | Treated Group | Control Group | Difference | *t*-Statistics | Critical Value of Gamma ($\Gamma$) | Balancing Test (%bias) |
|---|---|---|---|---|---|---|
|  | (*N* = 3169) | (*N* = 2797) |  |  |  |  |
| Food | 0.58 | 0.59 | −0.01 | 2.22 ** | 1.05–1.10 | −6.9 |
| Non-food | 0.14 | 0.13 | 0.01 | 4.17 *** | 1.05–1.10 | 12.4 |
| Education | 0.06 | 0.05 | 0.01 | 3.47 *** | 1.10–1.25 | 10.2 |
| Health | 0.06 | 0.05 | 0.01 | 2.3 *** | 1.85–2.00 | 7.1 |
| Durable goods | 0.03 | 0.03 | 0 | 1.63 | 1.15–1.35 | 6.1 |
| Others | 0.14 | 0.15 | −0.01 | 1.91 * | 1.35–1.45 | −5.9 |

Note: *, ** and *** are significant at 10%, 5% and 1%, respectively. The treated group is remittance-receiving households, and the control group is non-remittance receiving households. Households that had a migrant member, but did not receive remittances are included as non-remittance recipient households, hence included under the control group.

The following section analyses the impact of total, internal, external and both internal and external remittances on household expenditure patterns. We analyzed the effects of remittances on different quintile groups and by ecological belts using separate matching procedures in each group in order to avoid duplications.

### 4.2.1. Total Remittances

Table 5 presents the results of the average treatment effects for both internal and external remittances combined with the total remittances, analyzing the treatment effects by per capita expenditure quintile and by ecological belts (in Table 6). The result shows that remittance income is more likely to increase the share of household budget allocated to the consumption of non-food and investment in education and health, while it tends to decrease for the share of food consumption and other goods. However, the share of household expenditure on durable goods, though remaining the same, is not statistically significant. The difference between treated and control groups is about one percent and positive for non-food, household human capital investment and health. However, it is negative for the share of food consumption expenditure. A marginal decline in the share of household food expenditure with remittances is in line with the result of Gobel (2013) and Adams and Cuecuecha (2010).

**Table 6.** Average treatment effects (total remittance) by per capita expenditure quintile groups and ecological belt.

|  | Per Capita Expenditure Quintile Groups | | | | | Ecological Belts | | |
|---|---|---|---|---|---|---|---|---|
|  | 1st | 2nd | 3rd | 4th | 5th | Mountain | Hill | Tarai |
| Food | 0.01 | −0.01 | −0.002 | 0.01 | 0.01 | −0.034 ** | −0.01 | −0.01 |
| Nonfood | 0.004 | 0.01 | 0.01 ** | 0.01 ** | 0.001 | 0.004 | 0.01 ** | 0.01 *** |
| Education | 0.003 | 0.01 | 0.01 * | 0.01 | 0.002 | 0.01 | 0.01 * | 0.005 |
| Health | 0.03 ** | 0.02 | −0.001 | −0.001 | 0.003 | 0.01 | 0.02 ** | −0.001 |
| Durable goods | 0.003 * | 0.001 | 0.004 | 0.001 | −0.01 | 0.003 | −0.002 | 0.001 ** |
| Others | −0.01 * | −0.01 | −0.02 ** | −0.03 ** | −0.01 | 0.02 | −0.01 | −0.01 *** |

Note: *, ** and *** are significant at 10%, 5% and 1%, respectively.

In general, remittances seem to reduce the expenditure share of food and allocate more resources to other categories, such as non-food, human development investment and health. It is also interesting to note that remittance income is more likely to reduce the share of other expenditure, such as on rent, tobacco and garbage.

This study further investigates the impact of remittances on household expenditure patterns with reference to the per capita expenditure quintile groups and ecological belts with the aim of getting better insights of the expenditure behavior of remittance recipient households. The findings show a positive impact of remittances on non-food expenditure in the third and fourth quintiles, education expenditure in the third and fourth quintiles, health and durable goods expenditure in the first quintile and education expenditure in the third quintile. However, remittance income is more likely to decrease the share of expenditure on other commodities (e.g., tobacco, electricity, rent, etc.) in the first, third and fourth quintiles. Looking through the ecological belts, remittance recipient households are likely to spend more on non-food in the hills and the Tarai, education and health in the hills and durable goods in the Tarai. However, the food expenditure share shows the tendency of a likely decrease with remittances in the mountains.

### 4.2.2. Internal Remittances

Internal remittances have a positive impact on food and health expenditure, and the difference between treated and control groups is about one percent for food and health expenditure (Table 7). It is interesting to mention that it has a negative impact on the expenditure of education, durable goods and others. Their strategies could be to survive first by spending on food and health and then invest on luxurious goods.

**Table 7.** Average treatment effect (internal remittance).

| | Budget Share | | Difference | Two-Way *t*-Test |
|---|---|---|---|---|
| | Treated Group (*N* = 1113) | Control Group (*N* = 2971) | | |
| Food | 0.58 | 0.57 | 0.01 | 2.91 ** |
| Nonfood | 0.138 | 0.139 | −0.001 | 0.40 |
| Education | 0.05 | 0.06 | −0.01 | 2.06 ** |
| Health | 0.07 | 0.06 | 0.01 | 1.92 * |
| Durable goods | 0.03 | 0.04 | −0.01 | 3.11 *** |
| Others | 0.16 | 0.17 | −0.01 | 2.95 ** |

Note: *, ** and *** are significant at 10%, 5% and 1%, respectively.

Unlike total remittances, internal remittances by per capita expenditure quintile groups and ecological belts show an increase in the expenditure on food consumption in the third and fifth quintiles, but this reduces the investment on education in the fifth quintile (Table 8). Remittance-receiving households in the fifth quintile are more likely to reduce the expenditure on durable and other goods. It is also interesting to say that internal remittances have a positive impact on the expenditure share of food in the hills, while on the contrary, remittance recipient households reduce the expenditure share of food in the mountains and the Tarai and allocate to increase the expenditure share in education in the mountains. Remittance recipient households in the hills are likely to reduce the expenditure share in durable and other goods by increasing expenditure share in education.

**Table 8.** Average treatment effects (domestic remittance) by per capita expenditure quintile groups and ecological belt.

| | Per Capita Expenditure Quintile | | | | | Ecological Belt | | |
|---|---|---|---|---|---|---|---|---|
| | 1st | 2nd | 3rd | 4th | 5th | Mountain | Hill | Tarai |
| Food | 0.004 | 0.02 ** | 0.01 | 0.003 | 0.014 *** | −0.08 *** | 0.05 *** | −0.02 * |
| Nonfood | 0.002 | −0.004 ** | −0.001 | −0.001 | −0.001 | 0.01 | 0.004 | −0.001 |
| Education | 0.001 | −0.004 | −0.003 | 0.001 | −0.003 ** | 0.011 ** | 0.02 *** | 0.003 |
| Health | −0.002 | 0.004 | 0.003 | 0.01 | 0.002 | 0.04 | 0.01 | 0.002 |
| Durable goods | −0.002 | −0.003 * | −0.002 | −0.001 | −0.002 * | 0.001 | −0.011 *** | 0.004 |
| Others | −0.003 | −0.01 | −0.01 | −0.002 | −0.01 ** | 0.01 | −0.012 *** | −0.001 |

Note: *, ** and *** are significant at 10%, 5% and 1%, respectively.

4.2.3. External Remittances

The impact of external remittances on the household expenditure pattern is positive and highly significant for the investment on health and durable goods (Table 9), while the share of expenditure on other goods is negative and statistically significant; however, other expenditures on food, non-food and education, though negative, are not statistically significant. This result is in line with studies concluding that external remittances have a significant impact on the accumulation of durable goods and non-food consumption especially on clothes and footwear, recreation and culture and utensils (Adams and Cuecuecha 2010; Viet Cuong and Mont 2012; Gobel 2013).

**Table 9.** Average treatment effect (external remittance).

| | Budget Share | | Difference | Two-Way *t*-Test |
|---|---|---|---|---|
| | Treated Group (*N* = 1459) | Control Group (*N* = 2971) | | |
| Food | 0.580 | 0.585 | −0.005 | 0.73 |
| Nonfood | 0.14 | 0.13 | 0.01 | 1.18 |
| Education | 0.06 | 0.07 | −0.01 | 0.69 |
| Health | 0.07 | 0.06 | 0.01 | 1.80 * |
| Durable goods | 0.04 | 0.03 | 0.01 | 4.11 *** |
| Others | 0.15 | 0.16 | −0.01 | 2.06 ** |

Note: *, ** and *** are significant at 10%, 5% and 1%, respectively.

More specifically, external remittances have a positive impact on the expenditure share of education and durable goods in the fifth, health in the second and others in the fifth quintiles (Table 10). However, remittances are likely to decrease food expenditure in the fifth quintile. Remittance recipient households tend to spend more on durable and other goods in the hills and durable goods in the Tarai, while remittances seem to increase the expenditure share of food in the mountains. In general, the expenditure on durable goods seems to be the more preferred area of investment of external remittances among the quintile groups and by ecological belts.

**Table 10.** Average treatment effects (external remittance) by per capita expenditure quintile groups and ecological belt.

| | Per capital Expenditure Quintile | | | | | Ecological Belt | | |
|---|---|---|---|---|---|---|---|---|
| | 1st | 2nd | 3rd | 4th | 5th | Mountain | Hill | Tarai |
| Food | 0.003 | 0.01 | 0.001 | 0.01 | −0.03 *** | 0.07 *** | −0.014 | −0.004 |
| Nonfood | 0.002 | −0.001 | −0.002 | 0.001 | −0.002 | −0.02 | −0.004 | 0.001 |
| Education | −0.001 | −0.003 | 0.002 | −0.002 | 0.01 *** | 0.01 | 0.004 | 0.001 |
| Health | 0.004 | 0.01 * | −0.004 | −0.001 | −0.003 | 0.01 | 0.003 | 0.002 |
| Durable goods | 0.004 | −0.001 | −0.001 | −0.001 | 0.004 *** | −0.004 | 0.011 *** | 0.004 * |
| Others | −0.003 | −0.01 | 0.001 | −0.003 | 0.02 *** | −0.013 | 0.01 * | 0.002 |

Note: *, ** and *** are significant at 10%, 5% and 1%, respectively.

4.2.4. Internal and External Remittances

The effect of both internal and external remittances on the household expenditure pattern is positive and significant for the investment on non-food, health and durable goods, while it has a negative impact on the expenditure of other goods, such as rent, fuel, electricity and tobacco (Table 11); however, other expenditures on food and education are not statistically significant. Remittance recipient households tend to spend more on health and non-food items, such as durable and luxury goods, as their income increases from both internal and external remittances.

**Table 11.** Average treatment effect (internal and external remittances).

| | Budget Share | | Difference | Two-Way *t*-Test |
|---|---|---|---|---|
| | Treated Group (*N* = 423) | Control Group (*N* = 2971) | | |
| Food | 0.59 | 0.58 | 0.01 | 1.32 |
| Nonfood | 0.15 | 0.13 | 0.02 | 2.83 *** |
| Education | 0.05 | 0.06 | −0.01 | 0.41 |
| Health | 0.08 | 0.06 | 0.02 | 2.51 ** |
| Durable goods | 0.04 | 0.03 | 0.01 | 2.02 ** |
| Others | 0.12 | 0.13 | −0.01 | 2.5 ** |

Note: *, ** and *** are significant at 10%, 5% and 1%, respectively.

Households receiving both internal and external remittances have a positive impact on the expenditure share of food in the third and fifth quintiles and health in the second, third and fourth quintiles (Table 12). However, remittances seem to have a negative impact on education expenditure in the second, third and fifth quintiles, non-food in the fifth quintile and durable and other goods in the third quintile. Remittance recipient households tend to spend more on non-food in the hills and durable goods in the Tarai, while remittance recipient households are more likely to decrease the share of expenditure on food in the Tarai and other goods in the hills.

**Table 12.** Average treatment effects (both internal and external remittance) by per capita expenditure quintile groups and ecological belt.

| | Per Capita Expenditure Quintile | | | | | Ecological Belt | | |
|---|---|---|---|---|---|---|---|---|
| | 1st | 2nd | 3rd | 4th | 5th | Mountain | Hill | Tarai |
| Food | 0.02 | 0.02 | 0.033 *** | 0.01 | 0.02 ** | 0.001 | 0.003 | −0.04 ** |
| Nonfood | 0.001 | −0.001 | −0.001 | 0.002 | −0.003 ** | 0.03 | 0.02 ** | 0.01 |
| Education | −0.01 | −0.01 * | −0.01 ** | −0.003 | −0.005 ** | −0.01 | −0.001 | −0.01 |
| Health | −0.01 | 0.014 *** | 0.01 * | 0.01* | −0.003 ** | 0.013 | 0.02 | 0.004 |
| Durable goods | −0.003 | 0.003 | −0.005 ** | −0.001 | −0.002 | 0.002 | −0.001 | 0.02 *** |
| Others | −0.01 | −0.01 | −0.02 *** | −0.001 | −0.01 | −0.004 | −0.01** | 0.003 |

Note: *, ** and *** are significant at 10%, 5% and 1% respectively.

The overall results, as well as disaggregation by internal and external remittances are all robust and significant in terms of the difference between recipient and non-recipient households. Remittances increase in the share of expenditure on non-food, health, education and durable goods and reduce in the share of food expenditure. The findings are all consistent among the per capita expenditure quintiles and by ecological belts with a few exceptions. For instance, internal remittances seem to increase the expenditure share of food in the second and fifth quintile groups, but not by the external remittances. On the contrary, external remittances are more likely to increase the expenditure share of health, durable and other goods in the fifth quintile. Our results indicate that there is evidence of a significant difference of the impact of internal and external remittances on household expenditure patterns. However, it is interesting to mention that remittances from both internal and external sources tend to be spent more on food and health and reduce the share of expenditure on education and durable goods. Among the ecological belts, remittances have a significant positive impact on the expenditure on non-food and human development investment, mainly in the hills and Tarai, as compared to the mountains.

## 5. Conclusions

A growing number of Nepali migrants working abroad has significantly contributed to the country's Gross National Product (GNP) and has also become an important source of foreign exchange

earnings. Furthermore, remittance income has also significantly contributed to maintaining household consumption and coping with economic shocks; it is therefore obvious interest among researchers and economists how recipient households' spending behavior of remittances by per capita expenditure quintile groups and ecological belts differs from each other. This study has examined the spending behavior of remittance recipient households among the different income groups residing in different ecological belts across the country.

The study has applied a propensity score matching technique to assess the effect of internal and external remittances on household expenditure patterns, using the data from a large sample survey of the NLSS conducted in 2010/2011 by the Central Bureau of Statistics, Nepal, with technical and financial support from the World Bank. The use of propensity scores allows designing and analyzing observational studies that enable reducing selection bias by matching different groups. In other words, this method is a way to correct the estimation of treatment effects controlling for the existence of these confounding factors.

A simple comparison of the average budget share of the remittance recipient and non-recipient households shows a marginal difference in the expenditure share on different consumption goods (e.g., food and nonfood) and investment (e.g., health and education). Remittance recipient households tend to spend more on consumption and human development investment in general, implying that remittance income appears to enable sustaining consumption. Looking at the motivation to remit, the probability of receiving remittances seems to be high if the household has a higher share of children under six years, a higher female to male ratio and living in a rural area with a high percentage of migrant people in the cluster; while on the other hand, receiving remittances is likely to be low if the household has a relatively large family size living in urban areas, a larger number of elderly people and engaged in non-agriculture activities, such as business, jobs and entrepreneurship. The results of the balancing tests such as the *t*-test for the equality of means in the treated and non-treated groups and the standardized bias before and after matching show well balanced covariates and effective matching in building a good control group. Likewise, the results of sensitivity analysis using Rosenbaum bounds show that the critical value of $\Gamma$ is mostly within the range of the majority of studies (e.g., 1.1–2.0), indicating that the effects of remittances on outcome variables are sensitive to hidden bias.

The evidence shows that households receiving remittances tend to spend more on non-food goods (e.g., apparel and personal care items, religious and social functions, etc.), with investment in health and durable goods (e.g., kitchen appliances, jewelry, furniture, motorcycle, car, electronic goods, etc.). The difference between treated and control groups is more than one percent. However, remittance recipient households tend to spend less on food consumption and other goods, such as fuel, electricity and rent. This result among per capita expenditure by quintile groups is found to be consistent showing a positive impact on non-food and durable goods both in expenditure quintile groups and the ecological belts with a few exceptions.

The findings based on the internal, external and both internal and external remittances indicate the evidence of a significant effect of remittances on the household expenditure patterns. Remittance income seems to reshape the household demands without any effect of the total income. More specifically, internal remittances relatively seem to stimulate the consumption of food and investment in health. External remittances are found to be more effective on the accumulation of durable goods as compared to other goods, while both internal and external remittances tend to increase the share of non-food and health expenditure. These findings are in line with the hypothesis that internal remittances often help consumption smoothing and the external remittances contribute to more expenditure on non-food and durable goods. An in-depth analysis shows that internal remittances appear to increase the expenditure on food consumption in the second and fifth quintiles and people residing in the hills, while external remittances show a positive effect on the expenditure on non-food, health and durable goods. Internal and external remittances appear to have positive impact of the expenditure share on food and health.

Remittance recipient households tend to spend more on some investment categories, such as durable goods, health and education, which are found to be considerably larger and consistent in internal, external and both internal and external remittance recipient households. However, there is evidence of spending more on non-food items and less on food consumption of the households receiving remittances. The results are somehow consistent with the findings of Edwards and Ureta (2003), Adams and Cuecuecha (2010), Córdova (2004), Viet Cuong and Mont (2012), Bohra-Mishra (2013) and Gobel (2013).

**Acknowledgments:** The authors would like to thank the Central Bureau of Statistics, Nepal, for providing data and allowing them to use for the analysis. Moreover, we acknowledge for the comments from different experts while the paper was presented in different forums. The views expressed in this paper are those of the authors, and do not necessarily represent those of the institutions they belong.

**Author Contributions:** The first and corresponding author, Sridhar Thapa, contributed the section of empirical approach, data and descriptive statistics, and empirical results. He was mainly responsible for the development of models, data analysis and interpretation of the results. The second author, Sanjaya Acharya, contributed the section of introduction and conclusion of the manuscript. He also assisted in preparing the manuscript and play an overall supervisory role for the research.

**Conflicts of Interest:** The authors declare no conflicts of interest.

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
