# Peer review of "Remittances and Household Expenditure in Nepal: Evidence from Cross-Section Data"

_economies, doi:10.3390/economies5020016_

Round 1

Reviewer 1 Report

This paper adds to the literature on the effect of remittances on household expenditures and to what extent it influences spending on investment versus consumption goods. The paper uses PSM as the main methodology thereby following other papers that analyzed the same question in other countries. Nepal is an interesting case due to its high level of migration and remittances. The paper is well structured and it is clear what the authors aim to achieve. However, the paper could be strengthened along the following lines:

Although the introduction refers to existing literature and elaborates the rationale for the study in Nepal, it lacks a theoretical framework guiding the analysis. Except for the reference to the permanent income hypothesis, the authors do not elaborate why households would spend remittances differently. Money is fungible, at least according to standard economic theory. This would mean that spending decisions are only affected by the level of income and not by the income source. Yet, more recent research shows that this claim may not always hold. Along the same line, the authors argue that it matters whether remittances are from internal or external migration, but they do not provide a rationale why this would be the case.

The empirical approach is straight forward and well justified. However, the authors should establish at an early stage what the treatment variables are and how they are defined/measured. The decision to use Kernel  matching should be further justified by robustness checks whereby other matching procedures are used. Given that no information on the matching performance and balance of the sample is provided, it is difficult to assess to what extent the PSM provides robust results. The authors may consider adding the results to the paper.

The section on data and descriptive statistics lacks a table with basic statistics on all variables used in the models. It is therefore difficult for the reader to put the findings into context. The analysis uses national survey data, but it remains unclear whether the authors apply the sample weights in their analysis.

Table 1: It is not clear what the unit of analysis is, individuals or households?  Are ‘beneficiaries’ all individuals living in a remittance recipient household or is it the percentage of households receiving? The average income, is it for households or per capita? If it is the former, it provides biased information as household size may differ considerably across location and along the welfare distribution. It would also be helpful if the authors indicate whether the differences are statistically significant.  

Table 2: The budget shares do not sum up to 1 as they should. Rather, for households with remittances the sum of shares is 1.093 and for those without remittances it is 1.037, which cannot be due to rounding. Given that spending also depends on the level of income (Engel Law for food, for example), it is suggested to include budget shares by expenditure quintile.

Empirical results: the authors include the percentage of migrant people in the sample population as explanatory variable (line 257), but do not explain, how this variable was created and at what level of detail (PSU? Community?).

Table 3: please add the base units in the case of categorical variables. Why does the number of observations differ if both internal and external remittances are considered? Please also use the same wording throughout the text (internal and external, or national and international).  The interpretation of the results in the table should be carefully reviewed in order to avoid unfunded statements.

Another major problem concerns the analysis of ATE by quintile groups. It is not clear how this has been done. In order to analyze the effects per quintile group, a separate matching procedure needs to be applied for each group. The same applies for the ecological belt. If the authors have done so, then this should be clearly noted in the methodology section, or in section 4. With respect to the analysis of the overall ATE in table 4a-1, the findings indicate that household income as such may be the determining factor. The authors claim that education (in the PSM) accounts for household living standard, but this is not a sufficient proxy, most probably. The authors should consider including other variables that may proxy household living standards. 

Author Response

The empirical approach is straight forward and well justified. However, the authors should establish at an early stage what the treatment variables are and how they are defined/measured. The decision to use Kernel matching should be further justified by robustness checks whereby other matching procedures are used. Given that no information on the matching performance and balance of the sample is provided, it is difficult to assess to what extent the PSM provides robust results. The authors may consider adding the results to the paper.

Response: Thank you for your observations and suggestions. Considering your suggestions, we have added few sentences to define treatment variables and justify further the use of Kernel matching (184-188). We also tested robustness using Rosenbaum bounds and added in Table 4a-1(365).

The section on data and descriptive statistics lacks a table with basic statistics on all variables used in the models. It is therefore difficult for the reader to put the findings into context. The analysis uses national survey data, but it remains unclear whether the authors apply the sample weights in their analysis.

Response: Yes, we have added table of descriptive statistics (summary table) but we did not analyse the sample weights.

Table 1: It is not clear what the unit of analysis is, individuals or households?  Are ‘beneficiaries’ all individuals living in a remittance recipient household or is it the percentage of households receiving? The average income, is it for households or per capita? If it is the former, it provides biased information as household size may differ considerably across location and along the welfare distribution. It would also be helpful if the authors indicate whether the differences are statistically significant.  

Response: Following your suggestions, instead of average household remittance income, we have used per capita remittance income. This is followed by the t-test for significance difference between average household remittance income and per capita remittance income (236). Result is incorporated in the text.

Table 2: The budget shares do not sum up to 1 as they should. Rather, for households with remittances the sum of shares is 1.093 and for those without remittances it is 1.037, which cannot be due to rounding. Given that spending also depends on the level of income (Engel Law for food, for example), it is suggested to include budget shares by expenditure quintile.

Response: In line your suggestions, we have checked data and did some changes but even that did not appear exactly 1. We did not remove this table because we put the results of Rosenbaum bound and balancing test in the same table. Table of the difference of budget shares by expenditure quintile is presented in Table 4a-2(375).

 Empirical results: the authors include the percentage of migrant people in the sample population as explanatory variable (line 257), but do not explain, how this variable was created and at what level of detail (PSU? Community?).

Response: Yes we have already added with more clarification of percentage of migrant people in the text (271).

Table 3: please add the base units in the case of categorical variables. Why does the number of observations differ if both internal and external remittances are considered? Please also use the same wording throughout the text (internal and external, or national and international).  The interpretation of the results in the table should be carefully reviewed in order to avoid unfunded statements.

Response: As the analysis is total remittances vs no remittances, internal remittances vs no remittances, external vs no remittances, and both internal and external vs no remittances, we conducted analysis separately and incorporated your comments and suggestions. For instance, if we estimate logit model for external remittance receiving households vs no-remittance receiving households, the total observations will be external remittance receiving households plus no-remittance receiving households.  Based on your suggestions, we use same wording (internal and external) throughout the text.

Another major problem concerns the analysis of ATE by quintile groups. It is not clear how this has been done. In order to analyze the effects per quintile group, a separate matching procedure needs to be applied for each group. The same applies for the ecological belt. If the authors have done so, then this should be clearly noted in the methodology section, or in section 4. With respect to the analysis of the overall ATE in table 4a-1, the findings indicate that household income as such may be the determining factor. The authors claim that education (in the PSM) accounts for household living standard, but this is not a sufficient proxy, most probably. The authors should consider including other variables that may proxy household living standards. 

Response: In line with your suggestions, we have explained in Section 4 about separate matching procedure for expenditure quintile group and ecological belt. We have also slightly changed the results of logit estimates because we use variable ‘female to male ratio’ instead of sex of household head (suggestions from another reviewer). In updated logit estimates, education does not seem highly significant and influential in the analysis. As you suggest, this could not seem as good proxy in the results. 

Reviewer 2 Report

The purpose of this paper is to analyze the impact of remittances on household expenditure patterns in Nepal using household data from the Nepal Living Standard Survey 2010/2011. The authors distinguish four types of household regarding the remittances receipt: those receiving domestic remittances; those receiving foreign remittances; those receiving both domestic and foreign remittances; those receiving no remittances (who are the reference group). The authors analyze the impact of those categories on four types of expenditure: food, education, health, durable goods and other expenses. The empirical methodology used in the article is the Propensity Score Matching Analysis. The authors conclude that remittance recipients spend more on the non-food items.

Major comments:

1. The author should define the treatment more clearly: the treated households are those who receive remittances from different source. It is not clear if the period consider is the last 12 months. Moreover, the authors do not specify whether treated and untreated households include migrant and non-migrant households. In fact, it is possible that the groups of recipient and migrant households overlap but do not coincide.

1. Some covariates used to estimate the propensity score are relate to the household head, but household headship is clearly endogenous to migration, so that it would be incorrect to rely on them in a “no remittances and no migration” scenario. For example, I suspect that recipient households are (disproportionately) female-headed. The authors match them with female-headed untreated households. Widowhood is closely related to female headship, so that you are comparing your treated households with a group of households whose expenditure choices could reflect this permanent negative shock. The choice of the variables is never justify in the article. See Bertoli and Marchetta 2014 for a discussion on covariates.

2. Following the previous comment, some descriptive statistics (by remittance status) of the variables selected for the propensity score should be included.

3. The PSM is based on the validity of the conditional indipendnece hypothesis that states that the selection in the treatment group is only base on observable characteristics. In other words, PSM methods do not address the selection bias linked to unobservable characteristics which could not be negligible. It is the reason why the author should test the sensitivity of matching estimates to the presence of unobservable characteristics. See Rosenbaum Analysis (Rosenbaum, 2002).

4. The interpretation of the estimates for the logit model (Table 3) should be better justified and compared to the extensive literature in the field.

References:

Rosenbaum, P.R. (2002). Observational studies, New York: Springer.

Bertoli, S., Marchetta, F. (2014). Migration, Remittances and Poverty in Ecuador. Journal of Development Studies, 50:8, 1067-1089.

Author Response

Major comments:

1. The author should define the treatment more clearly: the treated households are those who receive remittances from different source. It is not clear if the period consider is the last 12 months. Moreover, the authors do not specify whether treated and untreated households include migrant and non-migrant households. In fact, it is possible that the groups of recipient and migrant households overlap but do not coincide.

Response: Thank you for your suggestions and comments. Considering your suggestions and recommendations, we have added and explained the period of remittances (i.e. last 12 months) and treatment groups in relevant section (217). We also specify that the analysis considered only household receiving remittances as treated households and excluded the migrant household without receiving remittances.

1. Some covariates used to estimate the propensity score are relate to the household head, but household headship is clearly endogenous to migration, so that it would be incorrect to rely on them in a “no remittances and no migration” scenario. For example, I suspect that recipient households are (disproportionately) female-headed. The authors match them with female-headed untreated households. Widowhood is closely related to female headship, so that you are comparing your treated households with a group of households whose expenditure choices could reflect this permanent negative shock. The choice of the variables is never justify in the article. See Bertoli and Marchetta 2014 for a discussion on covariates.

Response: Article Bertoli and Marcheta (2014) was really helpful. As per your suggestions, we have excluded sex of household head and added new variable ‘female to male ratio’(211 & 482).

2. Following the previous comment, some descriptive statistics (by remittance status) of the variables selected for the propensity score should be included.                                                    

Response: In line with your suggestions, we have added table of descriptive statistics (summary table-211).

3. The PSM is based on the validity of the conditional indipendnece hypothesis that states that the selection in the treatment group is only base on observable characteristics. In other words, PSM methods do not address the selection bias linked to unobservable characteristics which could not be negligible. It is the reason why the author should test the sensitivity of matching estimates to the presence of unobservable characteristics. See Rosenbaum Analysis (Rosenbaum, 2002).                           

Response: Following with your suggestions, we did sensitivity analysis using Rosenbaum bound and put the results of critical value of gamma in Table 4a-2 (365).

4. The interpretation of the estimates for the logit model (Table 3) should be better justified and compared to the extensive literature in the field.

Response: Yes we made replacement of new variable -female to male ratio instead of sex of HH head- re-estimates the logit models (365). There are also few changes in results after re-estimating the logit models and the explanations are in light of the recent literature that appears in the field. 

Round 2

Reviewer 1 Report

The authors have incorporated most of the comments received, but there are some remaining issues and some new issues due to the additional tables included. 

1. The paper still lacks a theoretical framework for the analysis and it is weak in explaining why it is important to distinguish between internal and external remittances. As a result, the analysis does not elaborate why we see the differences in outcomes depending on whether external or internal remittances are concerned. 

2. The paper needs to be edited due to many language errors which makes some parts very hard to understand. 

3. Section 3.1 and table 1a: The descriptive statistics table is helpful, yet incomplete, confusing and in some cases simply inappropriate. The columns distinguish between 'migration' and 'no-migration'. In my view this should be remittances versus no remittances. Receiving remittances is not the same as having a migrant in a household as migrants not necessarily remit. Given that the focus of the paper is on remittances, the tables should compare remittance-recipient versus non-recipient households. Base categories for categorical variables need to be included as well (I have asked for those in my previous comments). For example education: it only includes primary, secondary, high school and university (as in the logit models), indicating that there must be a fifth category. The same applies to geographical/topological location and occupation. Categorical variables, like 'migration', need to be included separately for each category. Showing the average of a categorical variable is meaningless. The outcome variables should also be included in this overview (budget shares). The interpretation of the table (text above the table) is also erroneous, although this might be a language issue. The following sentence is illustrative: 'As shown in Table 1a, majority of migrants are male as the average sex of household head is 0.80 for no-migration and 0.60 for migration'. First of all, the table does not contain this variable (household head sex). More importantly, there is nothing like 'average sex'. The last sentence before table 1a is also rather incomprehensible. 

4. Table 1b and text explaining table: Line 217 starts with 'A migrant-sending household....'. This is irrelevant here as the table considers remittances and not whether the household has a migrant. The authors then tried to address my earlier comments that it would be good to know whether the differences in remittances per capita shown in table 1b are statistically significant across groups of households. They misunderstood. In Line 219-221 they say that the difference between household and per capita remittances are significant. One would hope so given that household level remittances are a manifold of remittances per capita. This whole sentence needs to be removed. The authors should also check that the numbers from the table quoted in the text matches the number presented in the table. 

Author Response

The authors have incorporated most of the comments received, but there are some remaining issues and some new issues due to the additional tables included. 

1. The paper still lacks a theoretical framework for the analysis and it is weak in explaining why it is important to distinguish between internal and external remittances. As a result, the analysis does not elaborate why we see the differences in outcomes depending on whether external or internal remittances are concerned. 

Thank you for your suggestions. In regard to your comments, we have incorporated why there is difference in receiving internal or external remittances in outcomes (line 89 to 99). We have also discussed in overall results (line 468-469) about internal and external remittances.

2. The paper needs to be edited due to many language errors which makes some parts very hard to understand. 

Response: Regarding your suggestions, we have adjusted and rephrased some texts and sentences mainly in the area you suggested.

3. Section 3.1 and table 1a: The descriptive statistics table is helpful, yet incomplete, confusing and in some cases simply inappropriate. The columns distinguish between 'migration' and 'no-migration'. In my view this should be remittances versus no remittances. Receiving remittances is not the same as having a migrant in a household as migrants not necessarily remit. Given that the focus of the paper is on remittances, the tables should compare remittance-recipient versus non-recipient households. Base categories for categorical variables need to be included as well (I have asked for those in my previous comments). For example education: it only includes primary, secondary, high school and university (as in the logit models), indicating that there must be a fifth category. The same applies to geographical/topological location and occupation. Categorical variables, like 'migration', need to be included separately for each category. Showing the average of a categorical variable is meaningless. The outcome variables should also be included in this overview (budget shares). The interpretation of the table (text above the table) is also erroneous, although this might be a language issue. The following sentence is illustrative: 'As shown in Table 1a, majority of migrants are male as the average sex of household head is 0.80 for no-migration and 0.60 for migration'. First of all, the table does not contain this variable (household head sex). More importantly, there is nothing like 'average sex'. The last sentence before table 1a is also rather incomprehensible. 

Concerning your comments, we have added categorical variables including budget share in the descriptive statistics (Table 1a). We have recalculated (mean and standard deviation) and presented as remittance recipient households vs non-remittance recipient households instead of migrant vs non-migrant. We have also completely rephrased some sentences in line with your suggestions.

4. Table 1b and text explaining table: Line 217 starts with 'A migrant-sending household....'. This is irrelevant here as the table considers remittances and not whether the household has a migrant. The authors then tried to address my earlier comments that it would be good to know whether the differences in remittances per capita shown in table 1b are statistically significant across groups of households. They misunderstood. In Line 219-221 they say that the difference between household and per capita remittances are significant. One would hope so given that household level remittances are a manifold of remittances per capita. This whole sentence needs to be removed. The authors should also check that the numbers from the table quoted in the text matches the number presented in the table. 

Response: Following your suggestions, we have changed line 217 and also removed the sentence as you suggested. We rearranged and rephrased some sentences. 

Reviewer 2 Report

In the definition of the treated and non-treated groups, it is not clear to me whether those migrant households who do not receive remittances are dropped from the analysis or included in the non-treated group. 

The new variable "female to male ratio" should include the migrants. I advice the author(s) to state it clearly.

References should be in alphabetical order

Author Response

In the definition of the treated and non-treated groups, it is not clear to me whether those migrant households who do not receive remittances are dropped from the analysis or included in the non-treated group. 

Response: Thank you for your suggestions. Concerning you suggestions, we have included those households which have absentees but did not receive any remittances, in control group. See line 380-381.

The new variable "female to male ratio" should include the migrants. I advice the author(s) to state it clearly.

Response: Following you suggestion, we have included migrant members in female to male ratio.

References should be in alphabetical order

Response: Regarding your suggestions, we have prepared and arranged References based on the author’s guideline/style. If it is fine with editorial board, we will put in alphabetical order. 

Round 3

Reviewer 1 Report

From a technical perspective, the paper is now okay and can be published. However, I strongly advise that a native English speaker/editor will review and revise the paper.